# Australian Veterinarians’ Perceptions Regarding the Zoonotic Potential of *Mycobacterium avium* Subspecies *Paratuberculosis*

**DOI:** 10.3390/vetsci7010033

**Published:** 2020-03-19

**Authors:** Kamal R. Acharya, Karren M. Plain, Richard J. Whittington, Navneet K. Dhand

**Affiliations:** Sydney School of Veterinary Science, Faculty of Science, The University of Sydney, 425 Werombi Road, Camden, NSW 2570, Australia; kach8080@uni.sydney.edu.au (K.R.A.); karren.plain@sydney.edu.au (K.M.P.); richard.whittington@sydney.edu.au (R.J.W.)

**Keywords:** *Mycobacterium avium* subspecies *paratuberculosis*, perception, Australia, veterinarians, zoonotic disease, precautionary principle

## Abstract

Public concerns over exposure to *Mycobacterium avium* subspecies *paratuberculosis* (MAP) or MAP components via foods of animal origin could have negative trade consequences, despite the absence of conclusive scientific evidence of a causal association between *Mycobacterium avium* subspecies *paratuberculosis* (MAP) and Crohn’s disease (CD). This study was conducted among Australian veterinarians to understand (a) their perceptions regarding the role of MAP in the causation of CD (an ordinal outcome), and (b) their consideration of the adoption of the precautionary principle against Johne’s disease (JD; a binary outcome). Ordinal and binary logistic regression analyses were performed to evaluate the association of explanatory variables with the above outcomes, respectively. Almost one-third of the respondents (32.2%) considered that MAP was likely to be involved in the causation of CD whereas more than two-thirds (69.8%) agreed with the adoption of the precautionary principle against JD. Veterinarians who were concerned about exposure to and/or getting infected with MAP were more likely to consider MAP as a causative agent of CD (odds ratio: 7.63; 95% CI: 1.55, 37.63) and favor the adoption of the precautionary principle against JD (odds ratio: 6.20; 95% CI: 1.90, 20.25). Those perceiving MAP as a causative agent of CD were also more likely to favor the adoption of the precautionary principle against JD (odds ratio: 13.2; 95% CI: 1.26, 138.90). The results suggest that Australian veterinarians, particularly those who consider MAP as a causative agent of CD are concerned about exposure to MAP and favor the adoption of the precautionary principle against JD. These findings can be useful for animal health authorities for designing JD control programs and policies.

## 1. Introduction

*Mycobacterium avium* subspecies *paratuberculosis* (MAP) has a diverse host range including ruminants and primates [1,2]. In domesticated and wild ruminants, MAP causes Johne’s disease (JD), a chronic enteritis prevalent in most countries with commercial livestock farming [3,4]. In humans, MAP is suspected to cause Crohn’s disease (CD), a chronic enteritis that is increasing in the incidence in developed countries, particularly among the juvenile age group [5,6,7]. Human exposure to MAP can occur via milk, meat and their derived products including powdered milk and cheese [8,9,10,11,12,13,14,15,16,17,18] as MAP is naturally excreted in milk [19] and disseminated in meat [17]. Moreover, milk and meat could be contaminated with MAP following fecal contamination from animals shedding MAP [19,20,21,22]. Furthermore, humans can be exposed to MAP in the farm environment and via environmental contamination of municipal water [23,24].

The association of MAP in the causation of CD was suspected in 1913 by T K Dalziel when he noted the analogy between the pathological lesions and clinical findings of JD in animals and CD in humans [25,26]. Since then several studies have supported an association between MAP and CD [27,28,29,30,31,32,33,34,35] although some other researchers could not confirm this association [36,37]. MAP has been isolated from the intestine, blood and breast milk of CD patients, including children, confirming the potential for MAP to infect susceptible humans and suggesting a role for MAP in the causation of CD in humans [28,38,39,40,41,42]. MAP has also been shown to be capable of opportunistically infecting immunocompromised humans [43]. Although evidence for association comes from several research studies and meta-analyses, there is insufficient evidence to fulfill Koch’s postulates and Hill’s criteria on causation. Regardless, the presence of MAP in food for human consumption is a concern [32,44,45] and could impact trade of ruminant products in the future, with the degree of economic consequences depending on consumers’ or trading partners’ perceptions of the risk [46].

Public health concern was noted as one of the direct or indirect reasons for the initiation of current JD control programs in various countries [47]. Historically, public health concern was also a reason behind initiating the JD control program in Australia [48]. However, the present market-demand-driven control strategy of JD in Australia does not clarify if public health concerns are one of the reasons for the control of JD [49,50]. The present JD control strategy envisions that the stringency of JD control program adopted by a producer will be guided by the market-demand [49,50]. Thus, the direction of future JD control programs will be driven by a collective perception of consumers and other stakeholders regarding potential public health consideration of JD, which will affect the market access of a producer. 

Although, perceptions of experts working in the area of JD research on the public health concern of MAP have been investigated [44], information about perceptions of veterinarians, are unavailable. Veterinarians are the key stakeholders in the development, implementation and evaluation of JD control programs and are in a position to influence the perceptions of both producers [51] and policy makers. Moreover, their perceptions are likely to be developed based on a synthesis of their knowledge and experience of working with both producers and regulatory bodies. Disease control and management programs have benefitted in the past from understanding the perceptions of veterinarians on control, prevention and management of diseases [52,53,54,55,56]. However, there is no information about the perceptions of veterinarians on the public health concerns of MAP infection in ruminants. Therefore, this study was conducted to (a) understand the perceptions of livestock veterinarians in Australia regarding the zoonotic potential of MAP, and (b) assess whether veterinarians favor the adoption of a precautionary principle when considering the control of JD. 

## 2. Materials and Methods

This study was conducted as a cross-sectional study following approval by the Human Research Ethics Committee of the University of Sydney (approval number 2016/181).

### 2.1. Target and Study Population

Since JD mainly affects ruminants, veterinarians involved in cattle, sheep, goat and alpaca practice were the target population for this study. The study population was the veterinarians who attended two conferences of the cattle and sheep special interest groups of the Australian Veterinary Association, as well as the general membership of these groups.

### 2.2. Sample Size

A sample size of 196 was determined for this study. This sample size can estimate the prevalence of Australian veterinarians perceiving that MAP is a cause of CD with 95% confidence and 7% precision assuming that 50% of the Australian veterinarians believe that MAP could cause CD. However, the required sample size would be smaller if the expected proportion was lower or higher than 50%.

### 2.3. Questionnaire Development and Administration

A questionnaire was developed following standard guidelines and principles [57,58]. Relevant literature including the peer-reviewed literature and the policy documents available online [49,50] was studied to draft the questionnaire. The questionnaire was reviewed by identified subject matter experts (*n* = 7) including field veterinarians, government veterinarians and scientists working in the area of JD in Australia and was further refined following a piloting among field veterinarians and subject matter experts (*n* = 6).

The questionnaire comprised a total of 42 questions, including five open and 37 closed questions, and took approximately 20 min to complete. Space was provided in the questionnaire for the respondents to provide descriptive comments. The questionnaire had three major sections. The first section collected demographic information such as gender, education, training, work experience and laboratory usage with an aim to identify possible confounders and their influence on the outcomes. The second section focused on collecting information on JD management and control for cattle and sheep. The third section focused on understanding perceptions of veterinarians with regards to the zoonotic potential of MAP, their use of biosecurity measures, disinfectants and biosecurity procedures and their concerns regarding the presence of MAP in human food of animal origin and the possibility of being exposed to MAP. 

The survey was administered during the Australian Cattle Veterinarians (ACV) conference, Ayer’s Rock (Northern Territory) from 6–8 April 2016, and the Australian Sheep Veterinarians (ASV) conference, Dubbo (New South Wales) from 9–10 September 2016. Later, an online questionnaire was hosted on the SurveyMonkey platform and the link to the survey was provided to the members of ACV and ASV through their newsletters. Four veterinarians requested a hard copy of the questionnaire to complete after reading the newsletter. The survey remained open for approximately 10 weeks from 31 August to 15 November 2016.

### 2.4. Statistical Analyses

Data collation was done in Microsoft Excel. The statistical analyses were conducted using GenStat v 16.2.11713 (VSN International Ltd., Hemel Hempstead, UK) and the figures were prepared using GraphPad Prism 7 for Windows, Version 7.02 (GraphPad Software, Inc., La Jolla, CA, USA). 

#### 2.4.1. Explanatory Variables

The explanatory variables used for this study are summarized in Appendix A. The proportions of veterinarians’ clients with JD infected properties for different species were averaged to obtain the overall proportion of clients with JD infected properties. Similarly, the proportions of their clients who had initiated control programs or were intending to control JD in different species were also averaged. Respondents’ concerns regarding exposure to and getting infected with MAP were combined into a single index variable (Table 1). Similarly, their levels of agreement on the zoonotic nature of MAP were also combined into a single index (Table 1). Fifteen out of the 102 respondents completed only the first section of the survey that focused on the general information about the respondents. Thus, a decision was made to exclude these responses from statistical analysis.

#### 2.4.2. Outcome Variables

The outcome variables for the results reported in this manuscript were based on two questions. The first outcome was ‘veterinarians’ perception on MAP as a causative agent of Crohn’s disease’. This variable with 5 levels was collapsed into a three-level ordinal outcome (likely, neutral and unlikely). Consideration of veterinarians with regards to adoption of ‘precautionary principle against Johne’s disease’ was the second binary outcome.

#### 2.4.3. Descriptive Analyses

Categorical variables were described using proportions and compared with the outcome variables using contingency tables. Summary statistics and graphical summaries were created for quantitative variables.

#### 2.4.4. Univariable Analyses

The associations of the explanatory variables with the ordinal and binary outcome variables were assessed by univariable ordinal and binary logistic regression models, respectively. An explanatory variable with a likelihood-ratio chi-square *p*-value of <0.20 in univariable analysis was retained for further analysis. Linear associations of the continuous explanatory variable with log-odds of the outcome variable were assessed by plotting the parameter estimates of the categories against mid-points of the respective categories. In the absence of linear associations, the explanatory variables were categorized. Collinearity between the explanatory variables was assessed by calculating Spearman’s rank correlation coefficients. A pair of variables was considered to be collinear if they had a Spearman rank correlation coefficient of >0.7. If collinearity was detected, one of the pair of variables with a stronger association with the outcome variable was retained for further analysis. In addition, variables with more than 10% of missing values were excluded from the analysis. The excluded variables were retested in the final multivariable model. 

#### 2.4.5. Multivariable Analyses

The selected explanatory variables from the univariable analysis (*p*-value < 0.20) were used to construct the multivariable model using a manual forward stepwise approach. One of the variables that had more than 10% missing values was not used in the multivariable model building but was tested in the final model. The explanatory variables that showed statistically significant associations (*p*-value < 0.05) with the outcome variable were retained in the final model. Biologically plausible first order interactions were tested and retained in the final model if statistically significant. The variables likely to be potential confounders like the geography of work and gender [48,59] were inserted into the final model to assess their effects. Outliers and influential observations were identified using residual diagnostics by estimating leverage and Cook’s distance [60].

## 3. Results

### 3.1. Response Rate

A total of 102 responses were received: 35 (34.3%) from attendees at the ACV conference, 22 (21.6%) from the ASV conference and 45 (44.1%) were completed online or submitted via mail. The overall response rate of the survey was difficult to calculate due to the different modes of survey administration. However, at the ACV and ASV conferences, the response rate was 23% and 21% of the attendees, respectively. A final sample size of 102 was realized, which provided a precision of about 10%.

### 3.2. Demographics of the Respondents

Approximately one-third of the 102 respondents were female (32.4%). A majority of the respondents (46.1%, *n* = 47) had graduated between 1980 and 2000, followed by those graduating in 2000 and later (36.3%, *n* = 37). Most of the respondents (55.9%, *n* = 57) had no further formal postgraduation qualifications. Geographically, many of the respondents worked in NSW (38.2%, *n* = 39) or Victoria (30.4, *n* = 31). The respondents worked in many sectors including private practice, government, academia, consultancy, and diagnostic laboratories. The majority of the practice time of the respondents was dedicated to production animals and the client base of the majority of the veterinarians was rural (86.3%, *n* = 88). Detailed results are presented in Appendix A. 

### 3.3. Awareness and Preparedness of the Veterinarians with Regards to JD Control

More than half of the veterinarians (57.4%) had participated in at least one continuing education program on JD in the past three years. Apart from this, 63 (62.4%) respondents had also completed Market Assurance Program training, which is a pre-requisite to be involved in the assurance program for JD in Australia. Most of the respondents (64/82) were aware of the BJD review that was being undertaken by Animal Health Australia (Appendix A). 

### 3.4. Consultation Frequency of Veterinarians with Regard to JD

Less than a third of veterinarians reported being consulted ‘most of the time’ or ‘always’ by farmers regarding JD control and management on farm (29.1%, *n* = 25) and diagnosis of JD (17.3%, *n* = 15); the majority (53.5% and 61.0%) were only consulted ‘sometimes’ for these purposes (Appendix A). 

### 3.5. Perceptions Regarding MAP as a Causative Agent of Crohn’s Disease

Only about a third of the respondents (32.2%, *n* = 28) considered that MAP was likely to be involved in the causation of CD although there were 26 neutral responses. Table 2 shows the contingency tables of the assessed explanatory variables compared to veterinarians’ perceptions for MAP as a causative agent of Crohn’s disease. Eight explanatory variables were significantly associated with this outcome (*p* < 0.20) when assessed with univariable ordinal logistic regression (Table 3). These variables were related to the location of the clients, their state, their continuing education regarding JD, the type of laboratory used to diagnose JD, their exposure to confirmed JD-positive cases and the proportion of JD infected properties among their client list. Furthermore, their concern level for exposure to MAP and getting infected with MAP increased the odds of perceiving the role of MAP in the causation of CD. Collinearity was not noted among the significant explanatory variables.

The eight explanatory variables significant in univariable analyses (Table 3) were tested in a multivariable ordinal logistic model by using a manual forward stepwise technique, and only two explanatory variables were significant in the final model. Although not significant, gender and geography of work were forced into the model as they could be potential confounders. The results are summarized in the top panel of Table 4, which shows that veterinarians who were concerned about exposure to and/or getting infected with MAP were more likely to consider MAP as a causative agent of CD compared to those who had no/some concerns. Likewise, veterinarians with moderate and higher proportions of JD-positive farms among their clients were more likely to consider MAP as a causative agent of CD compared to those with low proportions of JD-positive farms among their clients. No outliers and influential observations were identified.

### 3.6. Consideration for the Adoption of the Precautionary Principle for Johne’s Disease

More than two-thirds of the respondents (69.8%) considered that the ‘precautionary principle’ should be adopted in relation to JD. Univariable logistic regression analyses were performed to test the association of the explanatory variables with the outcome. Of the 24 variables tested, 12 explanatory variables were significantly associated (*p*-value < 0.20) with the outcome (see Table 5 and Table 6). Generally, the importance placed on public health as a reason to control JD in cattle and sheep, awareness about OJD regulatory arrangements, and the proportion of JD infected client properties were significantly associated with the outcome. Likewise, variables related to the perception of MAP as a causative agent of CD, concern index on MAP exposure and infection and agreement index regarding MAP as a zoonotic agent were also significantly associated with perceptions of respondents regarding the adoption of the precautionary principle against JD. Collinearity was not noted among the significant explanatory variables. These 12 explanatory variables were used to perform multivariable logistic regression modeling. The final model results are presented in the second panel of Table 4. It was evident that veterinarians who were concerned or highly concerned regarding MAP exposure and infection were more likely to consider the adoption of the precautionary principle against JD compared to those with no/lesser concern, after adjusting for their perception on MAP as a causative agent of CD. Likewise, when the concern index on MAP exposure and infection was fixed (i.e., for participants with the same concern level), veterinarians who had a higher level of agreement on MAP as a causation of CD had higher odds of considering the adoption of the precautionary principle against JD, compared to those considering the role of MAP in the causation of CD as unlikely. No outliers or influential observations were identified.

## 4. Discussion

The JD control/management approach in Australia has recently undergone a major change towards deregulation [49]. The new approach mandates that farmers have the primary responsibility for JD control and places emphasis on risk management, adoption of regional biosecurity measures and the use of an animal health statement by sellers and buyers of livestock rather than on regulation, as was done previously for managing both BJD and OJD. Since JD is endemic in Australia [61], as in many other countries, the program envisages managing JD along with other endemic diseases using the tools of biosecurity and risk assessment. However, public health concerns regarding MAP are not present for many other endemic diseases. The objective of this study was to assess if veterinarians considered MAP to have zoonotic implications and hence considered whether or not the precautionary principle be adopted for JD control. This information was especially desirable considering the recent changes made in JD management strategies in Australia, such that the control/management options are driven by the market, with veterinarians as major stakeholders in providing on-farm advice to producers.

A majority of the veterinarians (71.8%) had concerns regarding the possibility of exposure to MAP and getting infected with MAP. The concern index comprised of five concern statements covering the exposure to humans via contaminated meat and milk, direct exposure via contact with animals and MAP contamination of food of animal origin that is intended for infants, old, pregnant women and immunocompromised people. This finding is not surprising in the context where there are concerns regarding the public health implications of MAP and exposure to MAP via food of animal origin, especially via dairy products, and meat and direct contact [44,45]. Similar perceptions were observed when a survey was conducted of subject matter experts working in the area of paratuberculosis [44]. Veterinarians who were highly concerned about exposure to MAP and getting infected with MAP were not only more likely to consider MAP as a causative agent of CD, but this concern was also reflected in their consideration of the adoption of the precautionary principle for JD.

The concerns of veterinarians are justifiable in the light of the current state of knowledge. It is evident that milk produced by animals in a JD-positive herd could potentially be contaminated with MAP, either via direct secretion into milk by animals affected with JD, or via contamination of milk by feces from clinically and subclinically affected animals; feces can contain copious amount of MAP [19,20,21,22,62,63,64,65]. Likewise, owing to the ability of MAP present in raw milk to withstand various processing procedures, MAP or MAP DNA has been detected in ready-to-eat milk derived foods, like infant formula, table milk, cheese and curd, from many parts of the world, albeit in varying proportions [8,9,10,11,12,13,14,15,16,18,66,67,68]. Furthermore, tissues derived from animals that are intended for human consumption can contain MAP, either through systemic infection and dissemination of MAP to sites distant from the gut or via fecal contamination during the dressing process at the abattoir [17,69,70,71,72,73]. Moreover, an animal can acquire MAP during transportation and while in holding pens, and shed MAP in their feces passively or acquire a high load of MAP contamination on skin, as suggested from one study that showed a higher load of MAP on the skin of animals at a slaughter house compared to that in their lymph nodes [74]. Consequently, during deskinning and evisceration of the carcass, contamination with fecal organisms [75], including pathogens in the feces, is unavoidable. Similarly, the likelihood of carcass cross-contamination at the abattoir cannot be ruled out. Although, the risk of exposure to MAP via meat is low [74], the concern of exposure is a valid one [76].

This study also showed that the proportion of JD-positive farms among the veterinarians’ clientele influenced their perceptions regarding the zoonotic importance of JD. For example, veterinarians who had a higher proportion of clients with JD infected properties were more likely to consider MAP as a causative agent of CD. Interestingly, veterinarians who had a moderate proportion of clients with JD infected properties were more likely to consider the adoption of the precautionary principle against JD, whereas those having a high proportion of clients with JD infected farm were less likely to consider the adoption of the precautionary principle against JD, when compared to veterinarians having low proportion of clients with JD positive farms. This observation possibly could be explained by various factors that have been known to influence human perceptions, like structural or institutional factors (macro level factors), peer-to-peer or community level factors (meso level factors) apart from the individual psychological level factors [77], which were not collected in the survey. Perhaps loyalty to clients and avoidance of measures that might damage the farm business (like quarantine), or even the risk of a loss of a client due to recommendation of unpopular measures, might have swayed the perception of veterinarians who had a high proportion of clients whose farms were affected by JD.

In contrast to a previous survey where 44.8% of specialists considered MAP as a likely risk to human health, thereby ranking it as a moderate public health issue [44], only one-third of the veterinarians in this study (32.2%) considered a likely role of MAP in the causation of CD. However, more than two-third of the respondents, including most of the respondents who did not consider a likely role of MAP in the causation of CD, considered that the precautionary principle should be adopted against JD as a potential zoonotic disease. Although the potential zoonotic importance of JD is explicitly mentioned as a motive for control by some countries, others have been cautious in acknowledging this. There is no doubt that once a country, continent or a trading block establishes an absolute or practical freedom from JD, trade of livestock and livestock products would be affected due to the adoption of a precautionary principle in order to preserve the sanitary status of the animals and human health, within the premise of the World Trade Organization Agreement on Sanitary and Phytosanitary measures agreement [78]. At present, some countries have more stringent control measures than others. Therefore, varying levels of prevalence of JD and hence MAP contamination in the food of animal origin are expected. Consumers expect to be able to identify the country of origin of animal derived food and food ingredients [79]. The impact of this consumer awareness on trade with respect to MAP is an area worthy of investigation, with a study suggesting economic consequences when the consumer perception of risk is high [46].

The JD control strategy in Australia acknowledges veterinarians as major stakeholders [49,50]. The results of this study show that more than two-thirds of the veterinarians were consulted at least sometimes in the control, management and diagnosis activities of JD on the farm; however, only limited numbers were consulted often. The reason for veterinarians not being more involved might be due to the veterinarians’ and/or the farmers’ perceptions and beliefs. A study highlighted that the proportion of time a veterinarian proactively spends in an advisory role with regard to preventative measures on a farm might depend on their required skill and motivation level, as well as whether counseling was seen as an income opportunity by the veterinarians, and if the farmers sought counseling [80]. In a survey of British farmers, inconsistent service, high turnover and a lack of expertise of sheep farming and concerns about independence of the advice given, were identified as barriers against the proactive use of veterinarians in sheep flock health management [81]. Veterinarians may be more involved in dairy health management activities [82]. The same study [81] identified that farmers considered themselves more knowledgeable about their own farm owing to the complexity of sheep farming. It is, however, not known if this would hold true in the Australian setting. Frequent involvement of veterinarians would be beneficial in the management of not only JD but other endemic diseases. In order to meet the expectation of the prevalent control measures, the involvement of the veterinarians in JD management activities (control, management and diagnosis) on the farm should be enhanced. This could be achieved by greater involvement of veterinarians in continuing education programs and training activities, particularly regarding the expectations of the currently adopted control program in Australia.

This is the first systematic study conducted in any countries to assess the perceptions of veterinarians regarding the zoonotic potential of MAP. The findings from the study would be useful not only to guide JD control programs in Australia but would also be valuable for refining control programs internationally. The analyses were conducted using multivariable approaches, which enabled us to evaluate associations after adjusting for confounders and other variables. Both geography of work and gender were identified a priori as potential confounders, because the prevalence of JD is different in various states of Australia [48] and gender differences with regard to risk perception have been noted in other studies measuring perceptions [59]; therefore these factors were forced into the final model irrespective of whether they were significant associations identified in the univariate model.

A limitation was the sample size achieved in the study, which was smaller than expected and thus the confidence intervals for proportions would be wider than expected. We attempted to improve participation by distribution of the questionnaire during two relevant national conferences. Moreover, the opportunity was provided to conference attendees to return the completed questionnaire by mail after the conference. Further, the survey link was posted in the newsletters of the ASV and ACV. Despite these efforts to increase the response rate, we acknowledge that the veterinarians participating in this survey may not be representative of all veterinarians working with food animals in Australia. This could introduce bias in the study, if the study sample (respondents) was systematically different from the rest of the population (non-respondents). However, the participants had a wide range of work experiences: working in private practice, academia, laboratories, government organizations and research and consulting activities. Participants had experiences ranging from less than 5 years to more than 40 years. Geographically, the participants had worked in all of the states of Australia and with the major species of livestock susceptible to MAP. Therefore, we believe that, if present, selection bias is not likely to be substantial [83]. Similarly, potential bias through two different modes of survey administration is possible; this was tested by including this variable in the model and it was found not to be significant. Confounding bias in this study was addressed by incorporating the variables identified a priori as potential confounders.

Furthermore, missing data are unavoidable in surveys and could be a source of bias. There are various approaches to deal with missing data depending on the source. In this study the missing data source was a complete nonresponse, where 15 respondents failed to provide any response for the explanatory and outcome variables. Imputation could be an option to address this, however, the precision of the estimate might be overestimated by that approach. Thus, it was decided to perform a list-wise deletion, as the nonresponse was assumed to be completely random. However, we acknowledged that the missing data might not be random and might have introduced systemic bias in the study.

Although, the perceptions of other stakeholders like consumers, producers and regulatory bodies are not known, the perceptions of veterinarians assessed in this study might also reflect the perceptions of producers. This was evident from a study that showed agreement between the risk perceptions of farmers and that of veterinarians [84]. This could possibly be due to the influence of veterinarians on farmers, through the provision of information and consultation on various aspects of disease management, control and diagnosis. 

The results suggest that it is necessary to adopt efficient JD control measures. Although further research is essential to ascertain the causative relationship between MAP and CD, eventually consumers’ perceptions are likely to necessitate that measures be taken to decrease the bacterial load in the food of animal origin to decrease human exposure to MAP, even in the absence of proof of a causative relationship between MAP and CD.

## 5. Conclusions

Although MAP was considered to be a potential zoonotic agent by only a third of the veterinarians surveyed, more than two-thirds of the respondents agreed on the adoption of the precautionary principle against JD. There was a high level of concern among veterinarians with regards to the presence of MAP in food of animal origin, and this was shown to be a reason for their consideration of the adoption of the precautionary principle against JD. These findings could be used for refining JD disease control programs.

## Figures and Tables

**Table 1 vetsci-07-00033-t001:** Responses for five concern measures and five agreement measures that were combined to obtain two index measures in the study conducted in Australia in 2016.

Questions						
***A. Concern Index on MAP Exposure and Infection ^a^***						
What is your level of concern regarding the given situations?		Frequency (Percentage)				
		Not at all concerned	Slightly concerned	Somewhat concerned	Fairly concerned	Very much concerned
1	Meat from Johne’s disease positive herd entering the food chain	31 (36.5)	30 (35.3)	13 (15.3)	11 (12.9)	0 (0.0)
2	Milk from Johne’s disease positive herd entering the food chain	22 (25.9)	23 (27.1)	17 (20.0)	17 (20.0)	6 (7.1)
3	Possibility of farmers getting infected with MAP	28 (32.9)	23 (27.1)	19 (22.4)	12 (14.1)	3 (3.5)
4	Possibility of you getting infected with MAP	33 (38.8)	25 (29.4)	19 (22.4)	6 (7.1)	2 (2.4)
5	Presence of MAP in food intended for infants, old, pregnant and immunocompromised people	13 (15.3)	21 (24.7)	22 (25.9)	19 (22.4)	10 (11.8)
***B. Agreement Index on MAP as a Zoonotic Agent ^b^***						
Please score your level of agreement		Frequency (percentage)				
		Fully disagree	Disagree	Neutral	Agree	Fully agree
1	I have asked my friends and families to take precautions with regard to the presence of MAP in the food of animal origin	32 (36.8)	33 (37.9)	16 (18.4)	5 (5.7)	1 (1.1)
2	I consider routine testing of animal products like meat and milk products for the presence of MAP should be essential	1 (1.1)	25 (28.7)	27 (31.0)	21 (24.1)	13 (14.9)
3	There is no evidence of the involvement of MAP in causation of Crohn’s Disease	12 (13.8)	27 (31.0)	31 (35.6)	13 (14.9)	4 (4.6)
4	Treating MAP as a zoonotic organism is a bit over the top	6 (6.9)	25 (28.7)	31 (35.6)	18 (20.7)	7 (8.0)
5	I think some people are over-reacting with role of MAP in Crohn’s disease	6 (6.9)	22 (25.3)	27 (31.0)	25 (28.7)	7 (8.0)

^a^ Categories related to agreement and disagreement were collapsed into two (1 and 0) and the scores for all the questions were added to obtain a total score. The total scores of 0 to 2 were grouped as “no/some concerns” and the scores of 3 to 5 were grouped as “concerned/highly concerned”; ^b^ Categories related to agreement and disagreement were collapsed into three (agree, disagree and neutral). The double negative responses of the last three questions were flipped before creating the index variable. The scores for all the questions were added to obtain a total score. Based on the score a respondent’s agreement index on *Mycobacterium avium* subspecies *paratuberculosis* (MAP) as a zoonotic agent was considered as “agree”, “neutral” and “disagree”.

**Table 2 vetsci-07-00033-t002:** Veterinarians’ perceptions for MAP as a causative agent of Crohn’s disease cross-tabulated against the assessed explanatory variables based on a study conducted in Australia in 2016.

Explanatory Variables	Categories	MAP as a Causative Agent of Crohn’s Disease			Total	N
		Likely	Neutral	Unlikely		
Mode of survey administration	ACV conference	9 (25.7)	7 (20.0)	19 (54.3)	35	87
	ASV conference	9 (41.0)	7 (31.8)	6 (27.3)	22	
	Others	10 (33.3)	12 (40.0)	8 (26.7)	30	
Year of graduation	Before 1980	6 (33.3)	5 (27.8)	7 (38.9)	18	87
	1980 to 2000	14 (36.9)	12 (31.6)	12 (31.6)	38	
	After 2000	8 (25.8)	9 (29.0)	14 (45.2)	31	
Gender	Female	9 (34.6)	7 (27.0)	10 (38.5)	26	86
	Male	18 (30.0)	19 (31.7)	23 (38.4)	60	
Level of education	Bachelor’s degree	15 (30.7)	16 (32.7)	18 (36.8)	49	87
	Higher Education	13 (34.3)	10 (26.4)	15 (39.5)	38	
Geography of work	NSW	11 (35.5)	9 (29.0)	11 (35.5)	31	87
	Victoria	5 (18.5)	8 (29.6)	14 (51.9)	27	
	Others	12 (41.4)	9 (31.1)	8 (27.6)	29	
Location of clients	Rural only	22 (29.0)	23 (30.3)	31 (40.8)	76	87
	Others	6 (54.6)	3 (27.3)	2 (18.2)	11	
Type of work	Private practice only	10 (29.4)	11 (32.4)	13 (38.2)	34	82
	No private practice	6 (54.6)	2 (18.2)	3 (27.3)	11	
	Other practice including private practice	11 (29.7)	12 (32.4)	14 (37.8)	37	
Animal worked with	Food animals	18 (34.6)	15 (28.9)	19 (36.5)	52	77
	Non-food animals	4 (28.6)	5 (35.7)	5 (35.7)	14	
	Both	3 (27.3)	4 (36.4)	4 (36.4)	11	
Laboratory use	Private only	8 (18.2)	16 (36.4)	20 (45.5)	44	81
	Others	17 (45.9)	8 (21.6)	12 (32.4)	37	
Continuing education on JD	Yes	20 (40.8)	12 (24.5)	17 (34.7)	49	86
	No	8 (21.6)	14 (37.8)	15 (40.6)	37	
Market assurance program training	Yes	16 (28.6)	17 (30.4)	23 (41.1)	56	86
	No	11 (36. 7)	9 (30.0)	10 (33.3)	30	
Consultation on JD control and management on farm	Yes	8 (32.0)	7 (28.0)	10 (40.0)	25	86
	Sometimes	14 (30.4)	15 (32.6)	17 (37.0)	46	
	No	5 (33.3)	4 (26.7)	6 (40.0)	15	
Consultation on JD diagnosis on farm	Yes	6 (40.0)	3 (20.0)	6 (40.0)	15	87
	Sometimes	17 (32.1)	16 (30.2)	20 (37.7)	53	
	No	5 (26.3)	7 (36.8)	7 (36.8)	19	
Diagnosis of JD ^a^	Single species	7 (24.1)	6 (20.7)	16 (55.2)	29	87
	Multi-species	17 (44.7)	12 (31.6)	9 (23.7)	38	
	None	4 (20.0)	8 (40.0)	8 (40.0)	20	
Awareness on BJD management programs in Australia	Agree	14 (34.1	8 (19.5)	19 (46.3)	41	80
	Neutral	6 (30.0)	8 (40.0)	6 (30.0)	20	
	Disagree	6 (31.6)	7 (36.8)	6 (31.6)	19	
Awareness on BJD review by Animal Health Australia	Yes	22 (34.4)	20 (31.3)	22 (34.4)	64	82
	No	6 (33.3)	3 (16.7)	9 (50.0)	18	
Awareness on OJD regulatory provisions	Agree	16 (44.4)	7 (19.4)	13 (36.1)	36	77
	Neutral	2 (11.8)	7 (41.2)	8 (47.1)	17	
	Disagree	7 (29.2)	10 (41.7)	7 (29.2)	24	
Concern index on MAP exposure and infection	Concerned/highly concerned	25 (41.0)	22 (36.1)	14 (23.0)	61	85
	No/some concern	3 (12.5)	4 (16.7)	17 (70.9)	24	
Know a CD patient	Yes	20 (33.9)	19 (32.2)	20 (33.9)	59	87
	No	8 (28.6)	7 (25.0)	13 (46.4)	28	
Updated knowledge on JD	Agree	16 (39.0)	6 (14.7)	19 (46.3)	41	87
	Neutral	6 (24.0)	12 (48.0)	7 (28.0)	25	
	Disagree	6 (28.6)	8 (38.1)	7 (33. 3)	21	
Proportion of clients with JD infected properties	High	6 (24.0)	5 (20.0)	14 (56.0)	25	74
	Medium	12 (50.0)	8 (33.3)	4 (16.7)	24	
	Low	4 (16.0)	11 (44.0)	10 (40.0)	25	
Proportions of clients who had initiated control programs or were intending to control JD	High	8 (36.4)	3 (13.6)	11 (50.0)	22	72
	Medium	5 (20.0)	10 (40.0)	10 (40.0)	25	
	Low	9 (36.0)	9 (36.0)	7 (28.0)	25	

^a^ Diagnosis made in a single species (bovine or ovine), multi-species (both bovine and ovine) and none. MAP: *Mycobacterium avium* subspecies *paratuberculosis*; ACV: Australian Cattle Veterinarians; ASV: Australian Sheep Veterinarians; NSW: New South Wales; JD: Johne’s disease; OJD: Ovine Johne’s disease; BJD: Bovine Johne’s disease.

**Table 3 vetsci-07-00033-t003:** Results of univariable ordinal logistic regression using ‘veterinarians’ perception for MAP as a causative agent of Crohn’s disease’ as an ordinal outcome based on a study conducted in Australia in 2016.

Explanatory Variable	Categories	Estimate	SE	OR (95% CI)	*p*-Value
Mode of survey administration	ACV conference	−0.83	0.47	0.43 (0.17, 1.09)	0.09
	ASV conference	0.16	0.52	1.18 (0.43, 3.25)	
	Others			1.00	
Geography of work	NSW	0.00		1.00	0.09
	Victoria	−0.75	0.50	0.47 (0.18, 1.25)	
	Others	0.31	0.48	1.36 (0.53, 3.46)	
Location of clients	Rural location only	−1.10	0.62	0.33 (0.10, 1.13)	0.07
	Others	0.00		1.00	
Laboratory use	Private only	−0.91	0.42	0.40 (0.18, 0.92)	0.03
	Others	0.00		1.00	
Continuing education on JD	Yes	0.54	0.41	1.72 (0.78, 3.79)	0.18
	No	0.00		1.00	
	Single species	−0.34	0.55	0.71 (0.24, 2.08)	0.03
Diagnosis of JD ^a^	Multispecies	0.90	0.52	2.44 (0.88, 6.74)	
	None	0.00		1.00	
Proportion of clients with JD infected properties	High	−0.31	0.53	0.73 (0.26, 2.08)	0.01
	Moderate	1.30	0.55	3.68 (1.25, 10.81)	
	Low	0.00		1.00	
Concern index on MAP exposure and infection	Concerned/ highly concerned	1.98	0.52	7.24 (2.63, 19.93)	<0.01
	No/less concerns	0.00		1.00	

^a^ Diagnosis made in a single species (bovine or ovine), multi-species (both bovine and ovine) and none; ACV: Australian Cattle Veterinarians; ASV: Australian Sheep Veterinarians; NSW: New South Wales; JD: Johne’s disease; MAP: *Mycobacterium avium* subspecies *paratuberculosis*.

**Table 4 vetsci-07-00033-t004:** Final ordinal logistic regression model for the outcome ‘veterinarians’ perception for MAP as a causative agent of Crohn’s disease, and the final binary logistic regression for the outcome ‘consideration of veterinarians with regards to adoption of ‘precautionary principle’ against Johne’s disease’ based on a study conducted in Australia in 2016.

Explanatory Variables	Categories	Estimate	SE	OR (95% CI)	*p*-Value
*Outcome 1: Veterinarians’ perception for MAP as a causative agent of Crohn’s disease*					
Constant (α1)		1.13	0.78		
Constant (α2)		2.97	0.85		
Concern index for MAP exposure and infection	Concerned/highly concerned	1.83	0.61	6.20 (1.90, 20.25)	<0.01
	No/some concerns	0.00		1.00	
Proportion of clients with JD infected properties	High	0.79	0.70	2.20 (0.57, 8.60)	0.03
	Medium	1.66	0.62	5.23 (1.57, 17.48)	
	Low	0.00		1.00	
Gender	Male	0.22	0.53	1.25 (0.45, 3.50)	0.69
	Female	0.00		1.00	
Geography of work	NSW	0.00		1.00	0.29
	Victoria	−1.12	0.70	0.33 (0.09, 1.30)	
	Others	−0.16	0.60	0.87 (0.27, 2.79)	
*Outcome 2: Consideration for the adoption of ‘precautionary principle’ against Johne’s disease*					
Constant (α)		−1.32	0.61		
Concern index on MAP exposure and infection	Concerned/highly concerned	2.04	0.82	7.63 (1.55, 37.63)	<0.01
	No/less concern	0.00		1.00	
MAP as a causative agent of CD	Likely	2.58	1.20	13.21 (1.26, 138.90)	0.04
	Neutral	0.37	0.79	1.44 (0.32, 6.68)	
	Unlikely	0.00		1.00	
Gender	Male	−0.72	0.86	0.50 (0.10, 2.65)	0.40
	Female	0.00		1.00	
Geography of work	NSW	0.00		1.00	0.66
	Victoria	−0.59	0.83	0.56 (0.11, 2.81)	
	Others	0.14	0.97	1.15 (0.18, 7.65)	

SE: standard error; OR: Odds ratio: CI: Confidence interval; MAP: *Mycobacterium avium* Subspecies *paratuberculosis*; JD: Johne’s disease; NSW: New South Wales; CD: Crohn’s disease.

**Table 5 vetsci-07-00033-t005:** Contingency tables of the ‘consideration of veterinarians for adoption of ‘precautionary principle’ against Johne’s disease (JD)’ with assessed explanatory variables based on a study conducted in Australia in 2016.

Explanatory Variables	Categories	Adoption of a Precautionary Principle Against JD		Total	N
		Yes	No		
Mode of survey administration	ACV conference	23 (65.7)	12 (34.3)	35	86
	ASV conference	17 (77.3)	5 (22.7)	22	
	Others	20 (69.0)	9 (31.0)	29	
Year of graduation	Before 1980	14 (77.8)	4 (22.2)	18	86
	1980 to 2000	22 (59.5)	15 (40.5)	37	
	After 2000	24 (77.4)	7 (22.6)	31	
Gender	Female	21 (80.8)	5 (19.2)	26	85
	Male	38 (64.4)	21 (35.6)	59	
Level of education	Bachelor’s degree	37 (75.5)	12 (24.5)	49	86
	Higher Education	23 (62.2)	14 (37.8)	37	
Geography of work	NSW	21 (70.0)	9 (30.0)	30	86
	Victoria	14 (51.9)	13 (48.1)	27	
	Others	25 (86.2)	4 (13.8)	29	
Location of clients	Rural only	50 (66.7)	25 (33.3)	75	86
	Others	10 (90.9)	1 (9.1)	11	
Type of work	Private practice only	27 (73.0)	10 (27.1)	37	81
	Other practice including private practice	24 (66.7)	12 (33.3)	36	
	No private practice	6 (75.0)	2 (25.0)	8	
Animal worked with	Food animals	34 (66.7)	17 (33.3)	51	76
	Non-food animals	10 (71.4)	4 (28.6)	14	
	Both	10 (90.9)	1 (9.1)	11	
Laboratory used	Private	29 (65.9)	15 (34.1)	44	80
	Others	27 (75.0)	9 (25.0)	36	
Continuing education on JD	Yes	35 (72.9)	13 (27.1)	48	85
	No	25 (67.6)	12 (32.4)	37	
Market assurance program training	Yes	37 (67.3)	18 (32.7)	55	85
	No	22 (73.3)	8 (26.7)	30	
Consultation on JD control and management on farm	Yes	17 (68.0)	8 (32.0)	25	85
	Sometimes	31 (68.9)	14 (31.1)	45	
	No	11 (73.3)	4 (26.7)	15	
Consultation on JD diagnosis on farm	Yes	11 (73.3)	4 (26.7)	15	86
	Sometimes	33 (63.5)	19 (36.5)	52	
	No	16 (84.2)	3 (15.8)	19	
Diagnosis of JD ^a^	Single species	16 (55.2)	13 (44.8)	29	86
	Multi-species	28 (75.7)	9 (24.3)	37	
	None	16 (80.0)	4 (20.0)	20	
Importance of public health reason to control JD in cattle and sheep	Important/very important	21 (84.0)	4 (16.0)	25	84
	Slightly/moderately important	26 (72.2)	10 (27.8)	36	
	Not at all important	11 (47.8)	12 (52.2)	23	
Awareness on BJD management programs in Australia	Agree	28 (70.0)	12 (30.0)	40	79
	Neutral	13 (65.0)	7 (35.0)	20	
	Disagree	15 (78.9)	4 (21.1)	19	
Awareness on BJD review by AHA	Yes	46 (73.0)	17 (27.0)	63	81
	No	12 (66.7)	6 (33.3)	18	
Awareness on OJD regulatory provisions	Agree	24 (68.6)	11 (31.4)	35	76
	Neutral	9 (52.9)	8 (47.1)	17	
	Disagree	21 (87.5)	3 (12.5)	24	
Concern index on MAP exposure and infection	Concerned/highly concerned	51 (83.6)	10 (16.4)	61	84
	No/some concern	8 (34.8)	15 (65.2)	23	
MAP as a causation of Crohn’s disease	Likely	26 (92.9)	2 (7.1)	28	86
	Neutral	19 (73.1)	7 (26.9)	26	
	Unlikely	15 (46.9)	17 (53.1)	32	
Know a CD patient	Yes	40 (69.0)	18 (31.0)	58	86
	No	20 (71.4)	8 (28.6)	28	
Updated knowledge on JD	Agree	25 (62.5)	15 (37.5)	40	86
	Neutral	19 (76.0)	6 (24.0)	25	
	Disagree	16 (76.2)	5 (23.8)	21	
Agreement Index on MAP as a zoonotic agent	Agree	31 (91.2)	3 (8.8)	34	86
	Neutral	8 (80.0)	2 (20.0)	10	
	Disagree	21 (50.0)	21 (50.0)	42	
Proportion of clients with JD infected properties	High	14 (56.0)	11 (44.0)	25	74
	Medium	21 (87.5)	3 (12.5)	24	
	Low	15 (60.0)	10 (40.0)	25	
Proportions of clients who had initiated control programs or were intending to control JD	High	14 (66.7)	7 (33.3)	21	71
	Medium	16 (64.0)	9 (36.0)	25	
	Low	19 (76.0)	6 (24.0)	25	

^a^ Diagnosis made in a single species (bovine or ovine), multi-species (both bovine and ovine) and none; ACV: Australian Cattle Veterinarians; ASV: Australian Sheep Veterinarians; NSW: New South Wales; JD: Johne’s disease; OJD: Ovine Johne’s disease; BJD: Bovine Johne’s disease; CD: Crohn’s disease; MAP: *Mycobacterium avium* subspecies *paratuberculosis*.

**Table 6 vetsci-07-00033-t006:** Results of univariable binomial logistic regression analyses using consideration of veterinarians for adoption of ‘precautionary principle against Johne’s disease’ as an outcome based on a study conducted in Australia in 2016.

Explanatory Variable	Categories	Estimate	SE	OR (95% CI)	*p*-Value
Year of graduation	Before 1980			1	0.20
	1980–2000	−0.87	0.66	0.42 (0.12, 1.52)	
	Above 2000	−0.02	0.71	0.98 (0.24, 3.95)	
Gender	Male	−0.84	0.57	0.43 (0.14, 1.31)	0.12
	Female			1	
Level of education	Higher education	−0.63	0.47	0.53 (0.21, 1.35)	0.18
	Bachelor’s degree			1	
Geography of work	NSW			1	0.02
	Victoria	−0.77	0.55	0.46 (0.16, 1.37)	
	Others	0.99	0.67	2.68 (0.73, 9.88)	
Location of clients	Rural location only	−1.61	1.08	0.20 (0.02, 1.65)	0.07
	Others			1	
Diagnosis of JD ^a^	Single species	−1.18	0.67	0.31 (0.08, 1.15)	0.11
	Multispecies	−0.25	0.68	0.78 (0.21, 2.93)	
	None			1	
Importance of public health reasons to control JD in cattle and sheep	Important/very important	1.75	0.69	5.73 (1.50, 21.97)	0.02
	Slightly/moderately important	1.04	0.56	2.84 (0.95, 8.48)	
	Not at all important			1	
Awareness regarding OJD regulatory arrangements	Agree	−1.17	0.71	0.31 (0.08, 1.25)	0.04
	Neutral	−1.83	0.78	0.16 (0.04, 0.74)	
	Disagree			1	
Proportion of JD infected clients	High	−0.16	0.57	0.85 (0.28, 2.61)	0.03
	Medium	1.54	0.74	4.67 (1.10, 19.90)	
	Low			1	
MAP as a causative agent of CD	Likely	2.69	0.81	14.73 (3.01, 72.18)	<0.01
	Neutral	1.12	0.57	3.08 (1.01, 9.34)	
	Unlikely			1	
Concern index on MAP exposure and infection	Concerned/highly concerned	2.26	0.56	9.56 (3.21, 28.48)	<0.01
	No/less concerns			1	
Agreement index on MAP as a zoonotic agent	Agree	2.33	0.68	10.33 (2.74, 39.00)	<0.01
	Neutral	1.39	0.85	4.00 (0.76, 21.11)	
	Disagree			1	

^a^ Diagnosis made in a single species (bovine or ovine), multi-species (both bovine and ovine) and none; NSW: New South Wales; JD: Johne’s disease; OJD: Ovine Johne’s disease; CD: Crohn’s disease; MAP: *Mycobacterium avium* subspecies *paratuberculosis*.

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
