# Peer review of "Australian Veterinarians’ Perceptions Regarding the Zoonotic Potential of Mycobacterium avium Subspecies Paratuberculosis"

_vetsci, 2020, doi:10.3390/vetsci7010033_

Round 1
Reviewer 1 Report
Thank you for considering the feedback. The modification presented is appropriate. Once again, thank you for providing a well-written and scientifically sound manuscript.
Reviewer 2 Report
Thankyou for making the revisions on this paper.
This manuscript is a resubmission of an earlier submission. The following is a list of the peer review reports and author responses from that submission.
Round 1
Reviewer 1 Report
This is a well-written paper. The language is very clear, the rationale for the study is well-established, and the methodology is properly explained. There are very few grammatical errors. I have only one suggestion, listed below. Thank you very much for the submission.
Line 100: Consider re-wording this paragraph to indicate that the selected sample size is larger than if the proportion of veterinarians who believe MAP causes CD is not 50%. This is a compelling point and would strengthen your statement more than beginning with an assumption.
Author Response
Thank you. The paragraph has been rephrased as follows:
A sample size of 196 was determined for this study. This sample size can estimate the prevalence of Australian veterinarians with the perception that MAP is a cause of CD with 95% confidence and 7% precision provided that 50% of the Australian veterinarians believe that MAP could cause CD. However, the required sample size would be smaller if the expected proportion was lower or higher than 50%.
Reviewer 2 Report
This reads as a piece of well-performed and justified research. The methods are comprehensive, the analysis rigorous and the discussion thorough. Limitations are perhaps overstated but it is great to see such good discussion by the authors. The work itself probably targets a limited readership but for those working with large animals this is an important issue, and knowledge and attitudes are clearly important in implementing any kind of public health strategy. My only minor suggestion would be to perhaps expand on the value of this work to ongoing policy/strategy reform. Given the results what might be the next stages – further education or to inform research funders of the need for determining a causal relationship between MAP and CD perhaps?
Author Response
Thank you. The paragraph has been revised as follows:
The results suggest that it is necessary to adopt efficient JD control measures. Although further research is essential to ascertain the causative relationship between MAP and CD, eventually consumers’ perceptions are likely to necessitate that measures be taken to decrease the bacterial load in the food of animal origin to decrease human exposure to MAP, even in the absence of proof of a causative relationship between MAP and CD.